# Arachidic Acid-Carrying Phosphatidylglycerol Lipids Statistically Mediate the Relationship Between Central Adiposity and Cognitive Function in Cognitively Unimpaired Older Adults

**DOI:** 10.3390/nu17213405

**Published:** 2025-10-29

**Authors:** Maria Kadyrov, Luke Whiley, Kirk I. Erickson, Belinda Brown, Elaine Holmes

**Affiliations:** 1Centre for Computational and Systems Medicine, Health Futures Institute, Murdoch University, Harry Perkins Building, Perth, WA 6150, Australia; luke.whiley@murdoch.edu.au; 2Centre for Healthy Ageing, Health Futures Institute, Murdoch University, Perth, WA 6150, Australia; belinda.brown@murdoch.edu.au; 3School of Health Sciences, University of Notre Dame, Perth, WA 6160, Australia; 4Curtin Medical School, Faculty of Health Sciences, Curtin University, Perth, WA 6102, Australia; 5Curtin Medical Research Institute (Curtin-MRI), Curtin University, Perth, WA 6102, Australia; 6Dementia Centre of Excellence, enAble Institute, Curtin University, Perth, WA 6102, Australia; 7AdventHealth Research Institute, Neuroscience Institute, Orlando, FL 32804, USA; kirk.erickson@adventhealth.com; 8Department of Psychology, University of Pittsburgh, Pittsburgh, PA 15260, USA; 9Section for Nutrition Research, Department of Metabolism, Digestion and Reproduction, Imperial College London, London SW7 2AZ, UK

**Keywords:** lipidomics, cognition, ageing, central adiposity, WGNCA, phosphatidylglycerol, arachidic acid, dementia, metabolism, lipids

## Abstract

**Background:** Central adiposity is a modifiable risk factor for age-related cognitive decline and has been linked to lipid dysregulation. However, the mechanisms underlying this relationship, particularly the role of plasma lipids at the species level, remain poorly understood. This study investigates whether lipids mediate the relationship between central adiposity and cognition in cognitively unimpaired older adults. **Methods:** Ninety-four cognitively normal older adults (*n* = 94, mean age 69.0 ± 5.0 years, 54% female) were included in this study. Cognitive composite scores were derived from z-standardised neuropsychological assessments, and central adiposity was measured using the waist–hip ratio (WHR). Lipidomic profiling identified 918 lipid species, which were clustered into modules of highly correlated lipids using a Weighted Gene Co-Expression Network Analysis (WGCNA). Modules associated with the WHR and cognition were identified via partial Spearman’s correlation analysis, followed by a mediation analysis. **Results:** Of the 39 lipid modules identified, 1 enriched with phosphatidylglycerol (PG) lipids containing an arachidic acid (20:0) sidechain was positively correlated with cognition (ρ = 0.32, FDR *p* < 0.05) and negatively correlated with the WHR (ρ = −0.43, FDR *p* < 0.001). Mediation analysis revealed that this arachidic acid-carrying PG lipid-enriched module mediated the WHR–cognition relationship, with individual species PG (20:0_16:1), PG (20:0_18:1), and PG (20:0_18:2) also contributing individually. **Conclusions:** Arachidic acid-carrying PG lipids statistically mediate the WHR–cognition relationship in cognitively unimpaired older adults. These findings suggest that adiposity-related lipid pathways are detectable in cognitively unimpaired older adults and may represent targets for early intervention to preserve cognitive health.

## 1. Introduction

With the population of older adults growing rapidly, age-related cognitive decline, both ‘normal’ and pathological (i.e., dementia), has become a serious public health concern [1]. Whilst normal age-related cognitive changes are generally more subtle than those seen in dementia, both can impact quality of life, reduce independence, and increase reliance on support networks and healthcare systems [2]. Reflecting the importance of early risk reduction, the Lancet Commission on Dementia Prevention, Intervention, and Care has identified 14 modifiable risk factors, which, if targeted, could reduce up to 40% of the dementia risk [3]. Many of these risk factors—such as physical inactivity, obesity, insulin resistance, hypertension, and high LDL cholesterol—are related to lifestyle factors and are characterised by metabolic changes, particularly in lipid metabolism [4]. Central adiposity (i.e., the accumulation of fat around internal organs) is especially relevant, as it is strongly linked to metabolic dysfunction, systemic inflammation, and oxidative stress, all of which further exacerbate the risk of cognitive decline even in individuals without clinical impairment [5]. Alterations in lipid metabolism are therefore likely to play a key role in linking central adiposity to cognitive outcomes, given the critical roles of lipids in maintaining both neurological and metabolic health independently. However, the mechanisms by which central adiposity influences cognition through changes in lipid metabolism are still not well understood.

Lipidomics, the comprehensive analysis of all lipids in a biological system, provides a powerful tool for investigating metabolism, inflammation, and disease. More recently, it has also been used to investigate age-related shifts in lipid metabolism and their potential associations with cognitive function [6] and central adiposity [7]. Lipids are critical to preserving normal brain health due to their role in maintaining the integrity and fluidity of cellular membranes [8]. Their functional roles include neurotransmission [9], inflammation [10], and protection against anti-oxidative damage [11]. The dysregulation of specific lipid pathways can have serious consequences for brain health and is associated with age-related neurodegenerative diseases such as Alzheimer’s disease (AD) [12,13,14]. In addition to studies in clinical cohorts (e.g., those with AD), it is also important to investigate the impact of ageing on lipid metabolism in cognitively unimpaired populations. Such research may reveal early targets for preventive strategies aimed at maintaining cognitive health in older adults. Whilst some studies have identified certain lipid species like plasmalogens [15] and sphingomyelins [16] as relevant to both cognitive and metabolic health, much research continues to focus on summed totals of broad lipid classes (e.g., LDL cholesterol or triglycerides). However, due to the highly correlated nature of lipids, studying these classes in isolation may overlook important interactions and systemic patterns. Additionally, lipid associations with phenotypes (e.g., cognition or central adiposity) are often investigated independently. This approach overlooks lipid–phenotype interactions at the systems level and the potential effects of lipids as mediators of clinical traits. As such, more advanced approaches, such as LC-MS lipidomics, are required to characterise the complexity of lipid–phenotype interactions.

While LC-MS lipidomics provides a broad coverage of the lipidome, it also introduces analytical challenges. The over-representation of some lipid classes (e.g., triglycerides) and strong correlations between structurally similar lipids can make it difficult to disentangle biologically meaningful associations. To address these limitations, network-based methods such as Weighted Gene Co-Expression Network Analysis (WGCNA) [17] can be applied. WGCNA groups highly correlated lipids into modules that can be linked to phenotypic traits, allowing for a more holistic understanding of those lipid–phenotype relationships. Whilst originally designed for gene co-expression data, WGCNA has recently gained popularity in lipidomics studies. This is due to its ability to reduce false positives, improve statistical power, and provide more biologically meaningful insights into lipid interactions with clinical traits. As such, this study applies WGCNA to explore associations of plasma lipids, cognition, and central adiposity and to investigate whether lipids mediate the connection between central adiposity and cognitive function in cognitively unimpaired older adults.

## 2. Materials and Methods

### 2.1. Participants

This cross-sectional analysis utilised data from 94 participants from the Intense Physical Activity and Cognition (IPAC) study [18], where a 6-month randomised–controlled trial was performed to observe the effects of a high- and moderate-intensity aerobic exercise intervention on cognitive function in older adults. For the purposes of this study, only the pre-intervention data were used to explore the relationships between lipids, central adiposity, and cognition independently of the exercise intervention. Participants were community-dwelling, older adults (aged 60–80) that were cognitively unimpaired as defined by a Mini-Mental State Examination (MMSE) score ≥ 26. The use of plasma samples and subsequent lipidomic data received ethical approval from the human research ethics committees at Murdoch University (protocol code 2020/096) and Edith Cowan University, and the study was registered with the Australian New Zealand Clinical Trials Registry (ACTRN12617000643370). All methods for the IPAC study have been previously reported in detail in an open access protocol paper [19].

### 2.2. Cognitive Assessments

The IPAC cognitive test battery has previously been described in detail [19]. Briefly, the battery included the CogState computerised battery [20], the California Verbal Learning Test (CVLT-II) [21], the Brief Visuospatial Memory Test (BVMT) [22], The National Institutes of Health—Executive Abilities: Measures and Instruments for Neurobehavioral Evaluation and Research (NIH-EXAMINER) [23], the Trail Making Test (TMT) [24], and the Wechsler Adult Intelligence Scale-III (WAIS-III) [25]. For this study, averaged z-scores obtained from raw test data were used to generate individual composite scores for global cognitive function and the subdomains of attention, delayed recall, episodic memory, executive function, learning and working memory. Tasks included for each cognitive composite score are available in Appendix A. Tasks that indicated better performance through a lower score were inversed (e.g., [speed score]* − 1).

### 2.3. Physical Assessments

The waist–hip ratio (WHR) was selected as the proxy measure of central adiposity in this study due to its more accurate representation of central fat distribution compared to other measures like the body mass index (BMI) [26]. Unlike the BMI, which does not distinguish between fat and lean mass, the WHR captures fat distribution and is more strongly associated with metabolic dysfunction, cardiovascular risk, and cognitive outcomes in older adults [27]. The WHR was assessed following established anthropometric procedures, including measurements of weight, height, waist circumference, and hip circumference.

### 2.4. Blood Samples

Fasted (10 h overnight) blood samples were collected in PAXgene tubes for APOE genotyping, and tubes containing lithium heparin were used to isolate plasma for lipidomic analysis. Plasma blood samples were centrifuged (4 °C for 10 min at 1300× *g*), and the resulting supernatant was aliquoted into cryotubes for storage at −80 °C until analysis. QIAamp DNA Blood Maxi Kits (Qiagen, Hilden, Germany) were used to extract DNA from whole blood prior to analysis using TaqMan genotyping assays (rs7412, assay ID: C____904973_10; rs429358, and assay ID: C___3084793_20) to create a categorical variable reflecting APOE ε4 allele status (i.e., APOE ε4 carriers vs. APOE ε4 non-carriers).

### 2.5. Lipidomics

#### 2.5.1. LC-QqQ-MS Analysis of Plasma Lipids

The lipidomic workflow employed in this study has been previously described [28] and is briefly reiterated here for reference. Targeted lipid profiling was performed by ultra-high-performance liquid chromatography–tandem mass spectrometry (UHPLC-MS/MS) on an ExionLCTM coupled to a 6500+ QTRAP LC-MS/MS system (SCIEX, Concord, CA, USA) using an established protocol. A stable isotope labelled internal reference standard mixture was prepared by combining the Internal Standards Kit for LipidyzerTM Platform (part number: 5040156) purchased from SCIEX (Framingham, MA, USA) with lysoPG 17:1 (part no: 858127C) and lysoPS 17:1 (part no: 858141C) from the SPLASH LipidoMIX (part no: 330707), which were purchased from Avanti Polar Lipids (Sigma-Aldrich, North Ryde, NSW, Australia) and diluted in isopropanol. Using a Biomek i5 Automated Liquid Handler (Beckman Coulter, Mount Waverly, VIC, Australia) for high-throughput liquid handling, 90 μL of the internal standards mixture was added to 10 μL of plasma and centrifuged at 4 °C. Then, 50 μL of supernatant was aliquoted into a 350 μL 96-well plate (Eppendorf, Macquarie Park, NSW, Australia). An aliquot of 5 μL was injected into an Acquity Premier BEH C18 1.7 μm, 2.1 × 100 mm (Waters Corp., Milford, MA, USA) reversed-phase column at 60 °C. The mobile phases comprised 10 nM ammonium acetate in a solution of water/acetonitrile/isopropanol (50/30/20 *v*/*v*/*v*) in mobile phase A and isopropanol/acetonitrile/water (90/9/1) in mobile phase B. The chromatographic run was performed at a flow rate of 0.4 mL/min with a total cycle time of 15 min. The elution gradient commenced at 10% B, increased to 45% B at 2.7 min, 53% B at 2.8 min, 60% B at 8 min, and 80% B at 8.1 min, where it was held until 11.5 min. The mobile phase then reached 100% B at 12 min before returning to 10% B at 13 min to re-equilibrate for the remaining 2 min. Using polarity switching, the 6500+ QTRAP LC-MS/MS system utilised electrospray ionisation. The mass spectrometer was set to a capillary voltage of 5500 V (positive ion mode), −4500 V (negative ion mode), and a temperature of 300 °C. The gas pressures were set at 20 psi (curtain), 40 psi (ion source 1), and 60 psi (ion source 2). Data were acquired using time-scheduled multiple reaction monitoring.

#### 2.5.2. Mass Spectrometry Data Integration

Raw spectra acquired using Analyst^®^ (v1.7.1, Sciex, Concord, CA, USA) underwent peak integration using Skyline (v21.1; MacCoss, Seattle, WA, USA) software [29]. Data pre-processing and filtering was then performed in R (version 4.3.1, http://www.r-project.org) in RStudio (version 1.4.1). First, all samples and lipid features with more than 50% missing values were excluded. For the remaining samples and lipid features, missing values were imputed using half the minimum detected value. Next, lipid features with relative standard deviations exceeding 30% in plasma quality control samples were removed. Finally, signal drift was corrected using a random forest algorithm implemented in the ‘statTarget’ (version 3.21) R package [30], with quality control samples serving as reference points.

### 2.6. Data Analysis

#### 2.6.1. Pre-Processing

Prior to analysis, missing raw cognitive test scores were imputed using a random forest algorithm from the missForest function (missForest v1.5) [31] to predict the values of missing data based on observed values within the data matrix. The lipidomic dataset was log transformed and scaled using zero-mean and unit variance normalisation. To determine the appropriate statistical tests for correlation analyses, univariate normality was assessed with shapiro.test (rstatix v0.7.2) [32], and homogeneity of variance was evaluated using leveneTest (car v3.1.2) [33].

#### 2.6.2. Associations of Cognition with Central Adiposity

Partial Spearman correlation analyses were conducted to examine the associations between WHR and all cognitive domains, controlling for age, sex, and APOE ε4 carrier status. This non-parametric approach was implemented using the pcor.test function (ppcor v1.1), which allows for partial correlation testing while adjusting for covariates. Domains that showed a significant association with WHR following Benjamini–Hochberg false discovery rate correction (FDR *p* < 0.05), applied using the p.adjust function in base R, were selected for downstream analyses. The resulting correlation coefficients and *p*-values were visualised as a heatmap using the ‘ggplot2’ package (v3.4.0).

#### 2.6.3. Weighted Gene-Co Expression Network Analysis

Weighted Gene Co-Expression Network Analysis (WGCNA) was performed to identify modules of co-expressed lipids using ‘WGCNA’ (v1.73) R package [17]. Outliers were first detected and removed using the goodSamplesGenes function to ensure the quality of the input data. The soft power threshold was determined using the pickSoftThreshold function. Twenty candidate powers were evaluated, and a power of 9 was selected as it provided the best balance of scale-free topology model fit (R^2^ > 0.80) and high mean connectivity (Appendix A). An adjacency matrix was constructed using Pearson’s correlation coefficient values between each normalised lipid via the adjacency function with the chosen soft power, and the topological overlap matrix (TOM) was calculated using a default unsigned network to account for connections involving both positive and negative correlations using the TOMsimilarity function. Lipids were hierarchically clustered to create a dendrogram from a dissimilarity matrix (1-TOM) using the hclust function with the average linkage method. To generate preliminary module assignments, the cutreeDynamic function was utilised (parameters: method = ‘tree’; deepSplit = 4, to enable identification of more detailed and refined modules; minClusterSize = 5). Module eigenvalues (MEs), representing the first principal component of each module, were calculated using the moduleEigengenes function. Similar modules were merged based on the dissimilarity of their eigenvalues (1 − correlation) using the mergeCloseModules function with a cut height of 0.2, corresponding to a correlation threshold of 0.8 (Appendix A). Finally, the module eigenvalues were recalculated and were randomly assigned a numeric label.

#### 2.6.4. Module–Phenotype Associations

Partial Spearman’s correlation analyses were performed to examine the relationship between the lipid modules, WHR, and cognition (with age, gender, and APOE ε4 carrier status included as covariates). Modules associated with WHR and cognition were then selected for bootstrapped non-parametric mediation analysis (*n* = 5000 samples) to assess whether the relationship between cognition and WHR is mediated through lipids. This was achieved using the ‘mediation’ (version 4.5.0) R package [34], with all variables transformed into z-scores to produce standardised beta coefficients prior to analysis and gender, age, and APOE ε4 allele carriage entered as covariates. This pipeline was then applied to individual lipids within the significant modules to identify key drivers of module behaviour.

## 3. Results

### 3.1. Participant Demographics

The study cohort included 94 participants with a mean age of 69.0 ± 5.0 years, of which 54% (*n* = 51) were female (Table 1). On average, the participants had 14.0 ± 2.3 years of education, and 26% (*n* = 24) carried at least one APOE ε4 allele. Participants had an average BMI of 25.8 ± 3.6 kg/m^2^, with 1% (*n* = 1) of participants categorised as underweight, 44% (*n* = 41) as normal weight, 41% (*n* = 39) overweight, and 14% as obese (*n* = 13) according to WHO BMI categories (<18.5 underweight; 18.5–24.9 normal; 25.0–29.9 overweight; ≥30 obese). The mean WHR was 0.9 ± 0.1; using sex-specific thresholds (men ≥ 1.00; women ≥ 0.85), 5% of men (*n* = 2) and 39% of women (*n* = 20) met the criteria for central adiposity, corresponding to 22% (*n* = 22) of the overall cohort.

### 3.2. Relationship Between Central Adiposity and Cognition

The partial Spearman correlation analysis revealed that the WHR was negatively correlated with global cognition (ρ = −0.23, *p* < 0.05; Figure 1). Positive intercorrelations were observed among the cognitive domains, particularly between episodic memory and delayed recall (ρ = 0.94) and between episodic memory and learning (ρ = 0.61). Among the cognitive domains assessed, only global cognition showed an association with the WHR.

### 3.3. WGCNA Lipid Modules

From the raw spectral data, 918 lipid features passed quality control and were selected for statistical analysis. Of these 918 lipid features, the WGCNA identified 39 modules of covarying lipids. One module (m14; 7% of all lipid features) contained all of the lipids that could not be clustered with any of the remaining 38 modules and thus was excluded from downstream analyses. The total numbers of lipids and subclass compositions of each module are highlighted in Figure 2, and the corresponding lipid features that comprise each module can be found in Appendix A.

### 3.4. Associations Between Modules and Phenotypes

The partial Spearman correlation analysis revealed several modules that were associated with either the WHR or global cognition and one module that was associated with both. Module m21, enriched with phosphatidylglycerol (PG) lipids that carry a 20:0 (arachidic acid) fatty acyl side chain, was positively correlated with global cognition (ρ = 0.32, FDR *p* < 0.05) and negatively correlated with the WHR (ρ = −0.43, FDR *p* < 0.001). Modules m12—enriched with phospholipids, predominantly phosphatidylcholines (PCs), carrying an 18:2 fatty acyl side chain—and m7, enriched with lysophosphatidylcholines (LPCs), showed a similar pattern; however, only the negative associations with the WHR remained significant after the FDR correction. These positive associations with cognition and the negative association with the WHR may be indicative of a more favourable lipid profile, both cognitively and metabolically. Two additional modules also showed a metabolically favourable pattern, demonstrating significant negative associations with the WHR. These include modules m34 (ρ = −0.34, FDR *p* < 0.05), enriched with ether-linked phospholipids such as plasmenyl-phosphatidylethanolamine (PE-P) and plasmanyl-phosphatidylethanolamine (PE-O), and m30 (ρ = −0.32, FDR *p* < 0.05), which is enriched with hexosylceramide (HexCer) lipid species. In contrast, two modules that showed less favourable patterns, i.e., positive associations with the WHR, were both enriched in triacylglycerols (TGs), which included modules m3 (ρ = 0.32, FDR *p* < 0.05)—which comprised TG lipids carrying a 20:3 or 20:4 fatty acyl side chain—and m2 (ρ = 0.38, FDR *p* < 0.01), which predominately contained saturated fatty acyl-carrying TGs. The associations between the lipid modules, global cognition, and the WHR are highlighted in Figure 3A, while lipid features that comprise each significant module are displayed in Figure 3B.

### 3.5. Mediation Analysis of Lipids, Central Adiposity, and Cognition

#### 3.5.1. Lipid Modules

To further investigate the mediating effect of module m21 on the relationship between the WHR and global cognition, a series of bootstrapped (*n* = 5000 samples) mediation analyses controlling for age, gender, and the APOE ε4 allele carriage were performed. Results showed an indirect (mediation) effect of module m21 on the relationship between the WHR and global cognition (ACME = −0.072, 95% CI [−0.148, −0.017], *p* = 0.006; Table 2). There was no direct effect observed between the WHR and global cognition (ADE = −0.075, 95% CI [−0.193, 0.042], *p* = 0.203), suggesting that the direct effect of the waist–hip ratio on global cognition was attenuated after accounting for module m21. The total effect, representing the overall association between the WHR and global cognition (i.e., the sum of the direct and indirect effects), was significant (total effect = −0.147, 95% CI [−0.263, −0.030], *p* = 0.016). Approximately 49% of the total effect was mediated by module m21 (proportion mediated = 0.49, 95% CI [0.099, 1.642], *p* = 0.022). These results support partial mediation, indicating that module m21 explains a substantial portion of the relationship between the waist–hip ratio and global cognition.

#### 3.5.2. Individual Lipid Species

To explore whether the mediation effect of module m21 on the WHR-cognition relationship was driven by individual lipid species or the module as a whole, additional mediation analyses on the individual lipids within the module were performed. Among the five PG lipids in this module, three—PG (20:0_16:1), PG (20:0_18:1), and PG (20:0_18:2)—demonstrated significant mediation effects (Table 3). Specifically, their average mediation effects were all statistically significant with estimated proportions of the total effect ranging from 33 to 40%. The average direct effects of these lipids were not significant, indicating no evidence of a direct association between the WHR and global cognition after the influence of these lipids was removed. Notably, total effects of the waist–hip ratio on global cognition remained significant across all models, reinforcing the overall associations.

## 4. Discussion

Using a lipidomic network approach, this study identified a PG-enriched module (m21) that was significantly associated with both higher global cognitive performance and lower waist–hip ratios in cognitively unimpaired older adults. The mediation analysis indicated that module m21 partially explained the waist–hip ratio and global cognition relationship, suggesting that PG species may act as intermediaries through which central adiposity influences cognitive health. Additional modules enriched with PC, LPC, PE-P, and HexCer species showed similar inverse associations with the waist–hip ratio and positive trends with global cognition, while TG-enriched modules correlated positively with the waist–hip ratio. Together, these findings indicate that lipid profiles linked to adiposity may have early implications for cognitive function, even in the absence of clinical impairment.

The mediating effect of PG-enriched module m21 on the inverse relationship between the waist–hip ratio and global cognition highlights a potential link between central adiposity and systemic lipid alterations. Specifically, a higher waist–hip ratio was associated with lower module eigenvalues, reflecting reduced concentrations of PG lipids comprising this module. This aligns with recent evidence that demonstrates the role PG lipids have in favourably modulating adipose tissue remodelling and inflammation [35], processes that are central to obesity-related metabolic dysfunction. Furthermore, this observed mediation by PG lipids is consistent with lipidomic analyses showing specific enrichment and altered levels of PG lipids in adipose-derived extracellular vesicles during obesity [36]. These vesicles have been identified as key mediators of adipose–brain communication, with evidence from high-fat diet models and patients with diabetes showing their contribution to insulin resistance, metabolic dysfunction, and cognitive decline. Together, these findings suggest a potential mechanism through which adipose-derived PG lipids may be transported to influence systemic metabolic and cognitive processes, thereby contributing to the observed relationship between the waist–hip ratio and global cognition. This study’s findings that m21 is inversely associated with the WHR suggests that a higher PG abundance may reflect a healthier metabolic phenotype, potentially buffering against adiposity-driven inflammatory stress.

The role of m21 in mediating the WHR–cognition relationship also aligns with the established functions of PGs in brain health. As precursors to cardiolipin—a phospholipid located in the inner mitochondrial membrane—PG lipids support mitochondrial integrity, bioenergetics, and mitophagy [36]. These are all processes that are often disrupted with ageing and are linked to cognitive decline. The dysregulation of PG metabolism can contribute to mitochondrial dysfunction, oxidative stress [37], and impaired neuronal function [38], providing a rationale for the positive association observed between PG lipids and global cognition. Although this study did not include individuals with neurodegenerative disease, it is notable that PG depletion has been reported in behavioural variant frontotemporal dementia [39] and AD progression [13]. Extending from this, experimental studies also show that PG supplementation restores mitochondrial function and synaptic activity in ABCA7 knockout models, a genetic risk factor for AD, underscoring PG’s potential to restore mitochondrial health and cognitive function [40]. In addition, PG exerts anti-inflammatory effects, including the inhibition of proinflammatory proteins such as toll-like receptors [41] and COX-2, suggesting that PG enrichment (as observed in m21) may counteract the chronic low-grade inflammation characteristic of central obesity and cognitive ageing. Remarkably, one study reported a noteworthy 358-fold inhibition of COX-2 mRNA expression with PG supplementation, highlighting its powerful capacity to regulate inflammation [42]. Given the central role of inflammation in obesity, ageing, and neurodegenerative diseases, this connection underscores the biological plausibility of PG’s mediating role in the relationship between the waist–hip ratio and cognition. Despite these promising findings, there remains a notable lack of research on PG in the context of cognition and healthy brain ageing, and further studies are warranted to explore PG’s therapeutic potential, particularly in obesity- and age-related cognitive decline.

Another notable feature of module m21 is that the PG species within this module contain an arachidic acid (20:0) fatty acyl chain, which may contribute to its prominent role as a mediator in the central adiposity–cognition relationship. Arachidic acid is typically obtained through the consumption of peanuts, peanut butter, and macadamia nuts [43] and can also be synthesised endogenously via the elongation of stearic acid (18:0). Positive associations between arachidic acid PGs and global cognition as reported in this study supports findings by Kotlega et al. [44], who described improved cognitive outcomes and greater neuroplasticity in stroke survivors with higher plasma arachidic acid levels. Similarly, Bi et al. [45] found reduced levels of arachidic acid in elderly adults with postoperative cognitive dysfunction, further supporting a potential neuroprotective role. Similarly, a recent study reported a negative association between arachidic acid and serum neurofilament light chain (NfL) concentrations, a biomarker of neuronal damage [46]. In contrast to other long-chain saturated fatty acids such as myristic acid (14:0), palmitic acid (16:0), and stearic acid (18:0), which have been linked to an increased risk of AD, arachidic acid does not appear to contribute to this risk [47]. Furthermore, Parilli-Moser et al. [48] found that a higher dietary intake of arachidic acid via peanut and peanut butter consumption was linked to enhanced memory, indicating the therapeutic potential of modifying levels of arachidic acid via lifestyle modifications. However, not all findings have been consistent. For example, Shen et al. [49] found no association between arachidic acid and cognition in older adults, whereas Dhillon et al. [50] reported elevated plasma levels of arachidic acid in individuals with mild cognitive impairment but not in AD or cognitively normal groups, suggesting a possible U-shaped relationship between arachidic acid and cognition. Overall, while several studies point to a supportive role of arachidic acid in cognitive health, these discrepancies highlight the complexity of its involvement in neurocognitive health and underscore the need for further research.

A growing body of evidence also suggests that arachidic acid may also influence cardiovascular and metabolic health. The negative association of arachidic acid PGs with the WHR found in this study is supported by Petersen et al. [51], who reported an inverse association of arachidic acid with BMI, waist circumference, and metabolic syndrome criteria. Consistently, most studies describe an inverse relationship between arachidic acid and cardiovascular disease or its risk factors [52,53,54,55,56,57], although a few have reported null associations [58] or adverse relationships in specific populations, such as in perinatally acquired HIV-positive children [59]. While the evidence overall supports a largely protective role of arachidic acid in cardiovascular health, findings for metabolic disorders such as type 2 diabetes are less consistent, with protective [43,60,61,62], null [63], and adverse [64,65] relationships reported. In a large prospective cohort study, higher arachidic acid concentrations were associated with a 17% lower risk of ‘unhealthy ageing’ (defined as any incident of CVD, severe kidney disease, chronic obstructive pulmonary disease, cancer, cognitive decline, or functional impairment) among participants without diabetes but not among those with diabetes [66]. This suggests that the beneficial effects of arachidic acid may be context-dependent and potentially attenuated in individuals with impaired metabolic function. Furthermore, a positive correlation between plasma leptin levels and arachidic acid has been reported, suggesting that arachidic acid levels may reflect greater adiposity or altered lipid metabolism linked to leptin signalling [67]. Given the overlap of risk factors for cardiometabolic diseases and cognitive decline, further investigation into the potential mechanistic relationship between arachidic acid, cognition, and metabolic regulation is warranted.

A limitation of our study is the sample size, creating several statistical challenges due to there being more variables than samples in the dataset. However, this was partially mitigated by using the WGCNA to reduce the dimensionality of the data and using multiple comparison methods to reduce the risk of false positives. Another limitation of this study is that we were unable to perform a separate validation in an independent cohort; hence, these results need to be further validated in a different cohort. Additionally, despite adjusting for an extensive list of covariates in our models, there is the possibility of residual confounding effects. However, it is unlikely that residual confounding could solely account for our results. We also included only cognitively unimpaired individuals in our study, limiting the observed cognitive range; hence, a stronger effect may be observed in a cohort that includes individuals who were cognitively impaired. Finally, whilst we observed a significant mediating effect of module m21 on the waist–hip ratio and global cognition, these results reflect statistical mediation not causal mediation. Therefore, longitudinal studies are required to confirm causality.

## 5. Conclusions

This study applied a lipid co-expression network analysis to examine plasma lipid associations with global cognition and the waist–hip ratio in cognitively unimpaired older adults. The analysis identified distinct lipid modules with opposing relationships to global cognition and the WHR, with one module (m21), enriched with arachidic acid-containing PG species, emerging as a potential mediator of the waist–hip ratio and global cognition relationship. Although participants were cognitively unimpaired, adiposity-related lipid signatures linked to cognitive function were detected, suggesting that these pathways arise early and may represent targets for preventing or delaying age-related cognitive decline through metabolic health interventions. The observed connections between PG species, mitochondrial function, and inflammation offer biologically plausible mechanisms by which adiposity influences brain health. These findings highlight the utility of lipidomic-based network approaches in uncovering biochemical pathways underlying cognitive ageing, although further external validation and longitudinal studies are needed to confirm causality.

## Figures and Tables

**Figure 1 nutrients-17-03405-f001:**
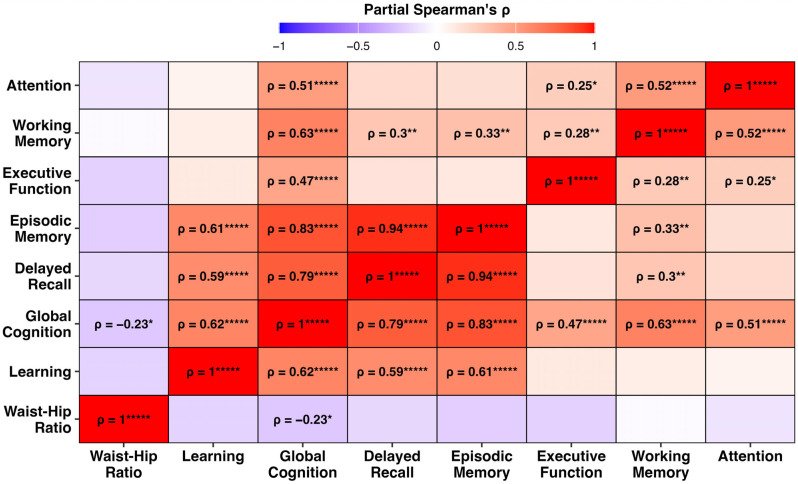
Heatmap of partial Spearman correlation coefficients (ρ) among waist–hip ratio (WHR) and seven cognitive domains, controlling for age, sex, and APOE ε4 carrier status. Warmer colours indicate stronger positive correlations, and cooler colours indicate negative associations. Only correlations that survived false discovery rate (FDR) correction are labelled (FDR *p* < 0.05). Asterisks denote significance levels: * *p* < 0.05, ** *p* < 0.01, and ***** *p* < 0.00001.

**Figure 2 nutrients-17-03405-f002:**
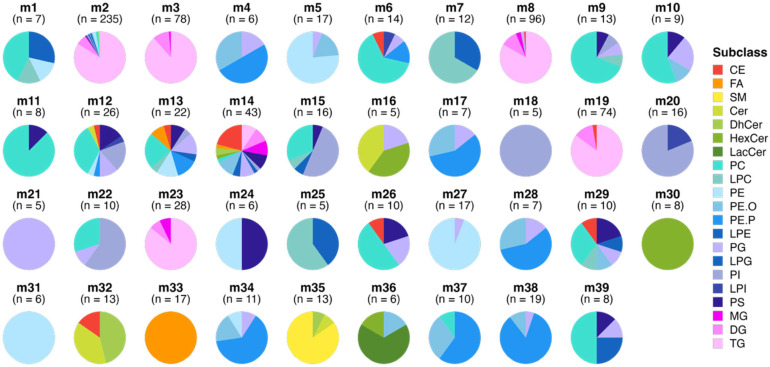
Lipid modules identified through Weighted Gene Co-Expression Network Analysis (WGCNA). Each pie chart represents a distinct module of co-expressed lipids, with the slices indicating the percentage of each lipid subclass within the module and n indicating the number of lipid species per module. The modules are ordered to highlight the enrichment of certain lipid subclasses and to illustrate similarities in the lipid composition between modules. Colours correspond to different lipid subclasses, demonstrating the variation and enrichment patterns across the lipid network. Module m12 represents all lipids that could not fit into any of the other modules. Abbreviations: CE, Cholesteryl Ester; Cer, Ceramide; DG, Diacylglycerol; DhCer, Dehydroxyceramide; FA, Free Fatty Acid; HexCer, Hexosylceramide; LacCer, Lactosylceramide; LPC, Lysophosphatidylcholine; LPE, Lysophosphatidylethanolamine; LPG, Lysophosphatidylglycerol; LPI, Lysophosphatidylinostol; MG, Monoacylglycerol; PC, Phosphatidylcholine; PE, Phosphatidylethanolamine; PE.P, Plasmenyl-PE; PE.O, Plasmanyl-PE; PG, Phosphatidylglycerol; PI, Phosphatidylinositol; PS, Phosphatidylserine; SM, Sphingomyelin; and TG, Triacylglycerol.

**Figure 3 nutrients-17-03405-f003:**
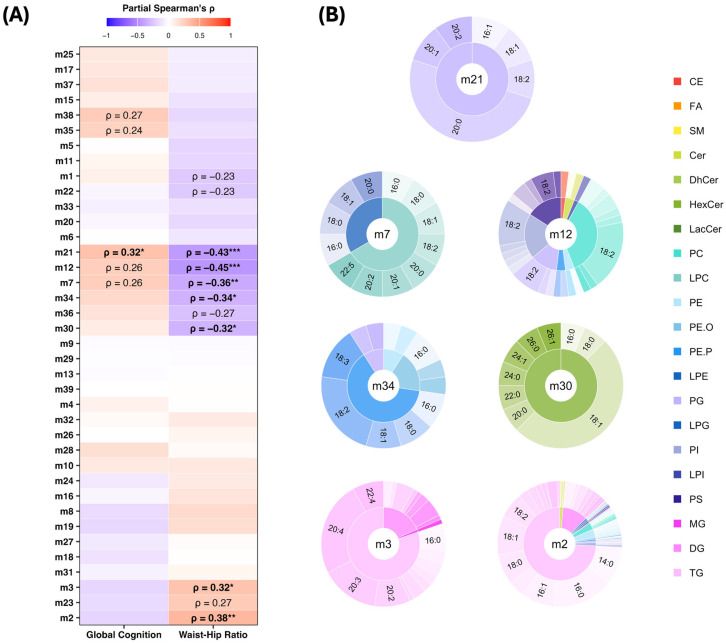
Associations of lipid modules with cognition and waist–hip ratio. (**A**) Partial Spearman correlation heatmap between lipid modules, cognitive composite scores, and waist–hip ratio. Text labels indicate significance of results (* *p* < 0.05, ** *p* < 0.01, *** *p* < 0.001), with associations that survived false discovery rate correction (FDR *p* < 0.05) highlighted in bold. (**B**) Pie–donut plots showing lipid class and sidechain composition of WGCNA modules significantly associated with global cognition and waist–hip ratio. Each module is represented by lipid subclasses in the inner ring, sidechains in the outer ring, and the module label in the centre. The sizes of the segments reflect the relative abundance of each lipid subclass and sidechain within the respective module. Colours correspond to lipid subclasses, and text labels indicate the most abundant lipid sidechains (top lipid species) within each module. Abbreviations: CE, Cholesteryl Ester; Cer, Ceramide; GC, Global Cognition Composite; PC, Phosphatidylcholine; PG, Phosphatidylglycerol; PE, Phosphatidylethanolamine; PE.P, Plasmenyl-PE; PE.O, Plasmanyl-PE; PS, Phosphatidylserine; ρ, Spearman’s Correlation Coefficient; SM, Sphingomyelin; and TG, Triacylglycerol.

**Table 1 nutrients-17-03405-t001:** Descriptive characteristics of study cohort.

Variable	*n* = 94 ^1^
Age, years	69.0 ± 5.0
Sex, female	51 (54%)
Years of Education	14.1 ± 2.3
APOE ε4 Allele Carriers	24 (26%)
Waist–Hip Ratio	0.9 ± 0.1
Men with central adiposity ^2^	2 (5%)
Women with central adiposity ^2^	20 (39%)
Total with central adiposity ^2^	22 (22%)
Body Mass Index (kg/m^2^)	25.8 ± 3.6
Underweight (<18.5)	1 (1%)
Normal weight (18.5–24.9)	41 (44%)
Overweight (25.0–29.9)	39 (41%)
Obese (>30)	13 (14%)
Cognitive Composites ^3^	
Attention	0.02 ± 0.73
Delayed recall	−0.07 ± 0.76
Episodic memory	−0.06 ± 0.75
Executive function	−0.03 ± 0.40
Global cognition	−0.03 ± 0.46
Learning	0.01 ± 0.70
Working memory	−0.02 ± 0.81

^1^ Mean ± SD for continuous; *n* (%) for categorical. ^2^ Central adiposity defined as WHR ≥ 1.00 for men and ≥0.85 for women (WHO criteria). ^3^ Cognitive composite scores represent z-standardised averages that follow an approximately normal distribution (mean = 0, SD = 1). Values near 0 indicate average cohort performance, with positive values reflecting above-average and negative values representing below-average performance.

**Table 2 nutrients-17-03405-t002:** Bootstrapped causal mediation analysis examining the indirect effect of waist–hip ratio on global cognition through module m21.

Term	Estimate	95% C.I.		*p* Value
		Lower	Upper	
Average Causal Mediation Effect (ACME)	−0.072	−0.148	−0.017	0.006
Average Direct Effect (ADE)	−0.075	−0.193	0.042	0.203
Total Effect	−0.147	−0.263	−0.028	0.016
Proportion Mediated	0.49	0.099	1.642	0.022

Bootstrapping was performed with 5000 resamples. ACME represents the indirect effect (i.e., how much of the effect of the waist–hip ratio on global cognition occurs through module m21), while ADE reflects the direct effect of the waist–hip ratio on global cognition after controlling for module 21. The total effect is the sum of the direct and indirect effects (Total Effect = ACME + ADE). The proportion mediated indicates the percentage of the total effect that is explained by the mediator (i.e., module m21).

**Table 3 nutrients-17-03405-t003:** Bootstrapped causal mediation analysis examining the indirect effect of waist–hip ratio on global cognition through individual lipids that comprise module m21.

Lipid	Effect	Estimate	95% C.I.		*p* Value
			Lower	Upper	
PG (20:0_16:1)	ACME	−0.745	−1.864	−0.165	0.004
	ADE	−1.239	−2.769	0.510	0.169
	Total Effect	−1.983	−3.580	−0.385	0.019
	Prop. Mediated	0.375	0.067	1.604	0.021
PG (20:0_18:1)	ACME	−0.661	−1.607	−0.044	0.030
	ADE	−1.323	−2.966	0.329	0.120
	Total Effect	−1.983	−3.565	−0.378	0.012
	Prop. Mediated	0.333	0.012	1.459	0.041
PG (20:0_18:2)	ACME	−0.797	−1.574	−0.174	0.008
	ADE	−1.186	−2.774	0.414	0.144
	Total Effect	−1.983	−3.536	−0.424	0.014
	Prop. Mediated	0.402	0.073	1.557	0.020
PG (20:0_20:1)	ACME	−0.380	−0.996	0.000	0.050
	ADE	−1.603	−3.005	−0.117	0.036
	Total Effect	−1.983	−3.511	−0.379	0.015
	Prop. Mediated	0.192	−0.008	0.680	0.062
PG (20:0_20:2)	ACME	−0.246	−0.973	0.295	0.410
	ADE	−1.737	−3.324	−0.068	0.041
	Total Effect	−1.983	−3.529	−0.336	0.022
	Prop. Mediated	0.124	−0.224	0.700	0.419

Bootstrapping was performed with 5000 resamples. ACME represents the indirect effect (i.e., how much of the effect of the waist–hip ratio on global cognition occurs through the lipid), while ADE reflects the direct effect of the waist–hip ratio on global cognition after accounting for the lipid. The total effect is the sum of the direct and indirect effects (Total Effect = ACME + ADE). The proportion mediated indicates the percentage of the total effect that is explained by the mediator (i.e., lipid).

## Data Availability

The original contributions presented in this study are included in the article/Appendix A. Further inquiries can be directed to the corresponding author.

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
