# Peer review of "Arachidic Acid-Carrying Phosphatidylglycerol Lipids Statistically Mediate the Relationship Between Central Adiposity and Cognitive Function in Cognitively Unimpaired Older Adults"

_nutrients, 2025, doi:10.3390/nu17213405_

Round 1

Reviewer 1 Report

Comments and Suggestions for Authors

Comments on nutrients-3920970

The aim of this study was to find a link between central adiposity and cognition in cognitively unimpaired older adults. 94 cognitively normal unimpaired older adults (aged 69±5 years – as mentioned in the abstract) were included in the study. These are still considered a younger group of elderly people. According to the title, I would have expected that a parallel group of elderly people with cognition issues ought also to be included for comparison. There is a high covariance in the study group. According to method section 2.1, the study group is heterogeneous with some performing high-intensity aerobic exercise, while others performing medium-intensity aerobic exercise. Are there any differences between these groups, the WHR and the lipid profile?

Title and abstract line 46: The verb "mediate" is not the proper word in this context. There is a correlation, but you cannot conclude that arachidic acid-carrying PG lipids "mediates". Please correct and tune down.

Line 48: How would you target arachidic acid-carrying PG lipids?

Introduction should add information on what is already known about arachidic acid and other blood fatty acids in cognition. E.g., doi: 10.1186/s12889-024-18478-x;  doi: 10.3390/ijms241814164; doi: 10.1016/j.arr.2020.101043; doi: 10.1002/alz.13792. 

Line 155: I think " LC-QQQ-MS" should be "LC-QqQ-MS" (liquid chromatographic-triple quadrupole tandem mass spectrometry). Please check.

Line 248: Since there are 94 participants, the n should be given for all the percentages.

The meaning of the numbers for cognitive composites should be described.

Line 273: Please correct to: "Asteriks".

Figure 2: The lipid modules should be described in more detail.  The meaning of n should be described in the legend. Is it the number of participants with the module?  M2 has a "n" of 235. How can it be, if there are 94 participants?  M21 – which contains phosphatidylglycerol (PG) lipids that carries the arachidic acid side chain has only a n of 5. Does it mean that only 5 of the participants expressed this variant? Or does it appear also in other modules?

How many of the participants were categorized with good cognition? The cutoff for positive/negative cognitive composites should be stated.

I think a Pie-Donut plot as presented in Figure 3 can be done also for Figure 2. This will be much more informative.

Line 302: should " hexosylcermaide" be " hexosylceramide"?

I couldn't find the Supplementary data.

Reviewer 2 Report

Comments and Suggestions for Authors

The authors present a lipidomic analysis based on data from 94 older adults, examining the relationship between waist-hip ratio (WHR), cognitive function, and lipid profiles. A lipid module containing phosphatidylglycerols with arachidonic acid residue (20:0) was identified, which mediates the relationship between central obesity and overall cognitive outcomes. A modern analytical approach—Weighted Gene Co-expression Network Analysis (WGCNA)—and mediation analyses were employed. This work makes a novel contribution to understanding the metabolic mechanisms of cognitive aging but requires several significant methodological and interpretative adjustments.

  1. Insufficient number of participants for analysis with 918 lipids
    WGCNA and mediation analyses with 94 participants and hundreds of variables pose a risk of overfitting and false positives. - this should be noted as limitation.
  2. No information is provided regarding the imputation and validation of WGCNA results.
    No cross-validation, modulus stability, or alternative segmentation (e.g., different soft-threshold values) is presented.
  3. Overly bold conclusions about a "mediating role"
    The article is based solely on a cross-sectional analysis. However, the authors suggest directionality and a potential mechanism of action.
  4. Lack of behavioral/metabolic data supporting a biological interpretation.
    Despite numerous references to the function of PG, measurements such as CRP, IL-6, HOMA-IR, SCFA levels, or dietary profile are missing.
  5. PGs containing arachidic acid are not unequivocally "beneficial."
    The authors assume that PG(20:0) is protective. However, the literature shows conflicting data regarding the role of saturated long-chain fatty acids.

Round 2

Reviewer 1 Report

Comments and Suggestions for Authors

The manuscript has been improved.

There is a problem in the pdf of the main manuscript for Figure 3 that was fallen out of the page.

Concerning Supplementary data - it would be preferable to put them into one document, and a more in depth figure legend description for the Supplementary data. 

Figure S1 lacks labeling of the axes.

Author Response

Comment 1: The manuscript has been improved.

Response 1: We thank the reviewer for their positive feedback and for recognising the improvements made to the manuscript.

Comment 2: There is a problem in the PDF of the main manuscript for Figure 3 that has fallen out of the page.

Response 2: We appreciate the reviewer drawing our attention to this formatting issue. The layout of Figure 3 has been corrected in the revised manuscript to ensure the entire figure and its legend are properly displayed within the page margins.

Comment 3: Concerning Supplementary data – it would be preferable to put them into one document, and a more in-depth figure legend description for the Supplementary data.

Response 3: All Supplementary Figures and Tables have now been consolidated into a single Supplementary Data word document. We have also expanded the figure legends to include more detailed methodological and interpretative descriptions for clarity and completeness.

Comment 4: Figure S1 lacks labeling of the axes.

Response 4: Axis labels have now been added to Figure S1 in the Supplementary Data to clearly indicate all variables presented.

Reviewer 2 Report

Comments and Suggestions for Authors

The Authors have responded to all Reviewer's suggestions. 

Author Response

Comment: The Authors have responded to all Reviewer's suggestions. 

Response: We thank the reviewer for acknowledging our responses and revisions. We appreciate their time and constructive feedback throughout the review process.